# Plant Growth-Defense Trade-Offs: Molecular Processes Leading to Physiological Changes

**DOI:** 10.3390/ijms22020693

**Published:** 2021-01-12

**Authors:** Juan Pablo Figueroa-Macías, Yamilet Coll García, María Núñez, Katy Díaz, Andres F. Olea, Luis Espinoza

**Affiliations:** 1Center for Natural Products Research, Faculty of Chemistry, University of Havana, Habana 10400, Cuba; jpfigueroa@estudiantes.fq.uh.cu; 2Departamento de Química, Universidad Técnica Federico Santa María, Avenida España 1680, Valparaíso 2340000, Chile; maria.nunezg@sansano.usm.cl (M.N.); katy.diaz@usm.cl (K.D.); 3Instituto de Ciencias Químicas Aplicadas, Facultad de Ingeniería, Universidad Autónoma de Chile, El Llano Subercaseaux 2801, Santiago 8900000, Chile; andres.olea@uautonoma.cl

**Keywords:** growth-defense trade-offs, phytohormones, cell receptors

## Abstract

In order to survive in a hostile habitat, plants have to manage the available resources to reach a delicate balance between development and defense processes, setting up what plant scientists call a trade-off. Most of these processes are basically responses to stimuli sensed by plant cell receptors and are influenced by the environmental features, which can incredibly modify such responses and even cause changes upon both molecular and phenotypic level. Therefore, significant differences can be detected between plants of the same species living in different environments. The comprehension of plant growth-defense trade-offs from the molecular basis to the phenotypic expression is one of the fundamentals for developing sustainable agriculture, so with this review we intend to contribute to the increasing of knowledge on this topic, which have a great importance for future development of agricultural crop production.

## 1. Introduction

Plants, as sessile organisms, are often facing various abiotic and biotic stresses that induce physiological, biochemical, and molecular changes, which are finally reflected in terms of lowest growth and productivity. Plant stress tolerance and susceptibility are governed by a complex exchange of signals and responses occurring at cellular and molecular levels. So, plant resistance relies strongly on timely stress recognition and the rapid and effective activation of defense mechanisms [1]. The response of plants is dynamic and complex and varies with the nature of these stresses, i.e., defense mechanisms for pathogen infection or drought stress conditions are different.

Plant immune responses are also modulated by abiotic signals such as light and temperature as well as the circadian clock [2]. In the absence of perceived pathogens or environmental changes, young tissues must suppress immune or adaptation responses to maximize growth, whereas mature organs can be more prepared for defense [3]. Thus, the activation of defense mechanisms at the expense of growth suspension is known as the “growth-defense trade-off” phenomenon (GDT) and is the result of a delicate balance between growth and defense or adaptation responses. Competition for limited amount of available resources has been considered as the driving force for trade-offs [4,5,6,7], but recently it has been proposed that instead this could be the result of opposite molecular pathways regulating growth and defense [8,9].

Due to climate change that is impacting negatively agricultural crops production worldwide, GDT have acquired increasing importance from the ecological, agricultural, and economic point of view. For centuries, agricultural crops have been bred with the main goal of maximizing growth and yields. As a consequence, plants have diminished its genetic diversity for abiotic stresses defense [10]. The knowledge about the molecular and regulatory aspects that manage the above-mentioned phenomenon is the very first step in the development of new agricultural products that stimulate both growth and defense responses. Thus, in this review, we intend to make an interesting approach to the GDT aimed to suggest elements for the analysis and better understanding of key pieces of knowledge involved in further development of agricultural applications.

## 2. Regulatory Aspects

Plants count on specialized mechanisms designed to detect and identify environmental changes, and subsequently produce the optimal response. Effective defense responses against both biotic and abiotic stresses are triggered by a series of signaling molecules, where phytohormones are the most important. Phytohormones, such as auxins (AUX), gibberellins (GAs), cytokinins (CKs), abscisic acid (ABA), ethylene (ET), brassinosteroids (BRs), salicylic acid (SA) and jasmonic acid (JA), are endogenous low-molecular-weight molecules, which in addition of their defense signaling function, are also regulators of growth, development, and a variety of physiological processes. In general, phytohormones act by modulating expression of a number of different genes whose products are so diverse as proteinases, cell-protection proteins, hydrolytic enzymes, pathogenesis-related proteins, reactive-oxygen species. Several of these responses to stress show synergistic or antagonistic effects and therefore these phytohormones should coordinate themselves through an efficient crosstalk to get the optimal response to the environmental changes and maintaining at the same time growth and development [11,12]. Understanding of mechanisms involved in GDT is important considering the urgent need of optimizing yields in agricultural crop species.

The role of these phytohormones in signaling mechanisms and crosstalk in response to biotic stresses have been recently reviewed [12]. However, changes in resources allocation induced by external pressures are quite difficult to determine. Thus, a working framework to study the costs of plant defense at the expenses of other plant’s functions has been proposed [13]. A common approach to study GDT is to detect and quantify the production of defensive chemicals induced by herbivory. Attack of plants by herbivores induces signals associated to wounding, damage-associated molecular patterns (DAMPs), and oral secretion, herbivory-associated molecular patterns (HAMPs) and fatty acid conjugates. These cues activate mitogen-activated protein kinases (MAPKs), which initiate signal transduction pathways, i.e., wound-induced protein kinases (WIPKs) are positive regulators of JA, and initiate cascades stimulating salicylic-acid-induced protein kinases (SIPKs) that additionally are agonistic of ET production and suppress SA synthesis. Thus, plants respond to attack of herbivores by triggering a signaling cascade to produce a variety of chemical defenses, and at the same time a series of metabolic changes that affect growth are observed. This change in resource allocation resulting in both increased defense and lowest rate of growth [14] requires a crosstalk between different phytohormones that often play opposite roles.

Nonetheless, signaling molecules are the “real regulators” behind hormones, since they are the ones that are activated upon perceptions and then, trigger the production or inhibition of hormones. Downstream of MAPKs, WIPKs, SIPKs, and transcription factor WRKY in the signal transduction network, they act to further enforce the regulatory crosstalk between growth and defense [13]. A case in point could be an experiment where oral secretions of caterpillars were applied to WIPK-silenced *N. attenuate*, treated plants displayed a less accumulation of JA—as WIPKs are agonistic regulators of JA—and consequently, they experienced higher growth and seed production relative to control plants. On the contrary, the opposite effect was observed in SIPK-silenced plants that underwent the same experiment, because plants accumulated high levels of SA and did not benefit from reduced JA sensitivity, presumably because of the higher costs associated with SA-mediated defense [14]. Another example is the transcription factor WRKY70 that is MAPK-activated as result of insect attack. It has been shown that WRKY70 acts to upregulate the lipoxygenase (LOX) gene involved in JA synthesis and concurrently inhibits the plant growth hormone gibberellin (GA) by suppressing the GA synthesis gene gibberellin 20–oxidase 7 (GA20ox7), resulting in a stunted phenotype [15]. From all that we could conclude that plants use most of growth and defense processes, which are regulated by these chemical messengers, and therefore phytohormones are who really determine whether the plant grow or defend itself.

Brassinosteroids, are an important class of phytohormones that can be found in almost all parts of the plants and play important role in a variety of biological processes, such as plant growth regulation [16,17] and cell division and differentiation in young tissues of growing plants [18,19,20]. It has also been shown that BRs increase resistance to various kinds of biotic and abiotic stress factors, i.e., low and high temperature, drought, heat, salinity, heavy metal toxicity, and pesticide [21,22,23,24,25,26]. Several studies have shown that BRs increase antioxidant enzymatic activity and reduce drought stress effects on plants as measured by different biochemical and physiological parameters, i.e., chlorophyll accumulation, activity of antioxidant enzymes, total protein contents, stomatal conductance, photosynthesis, and membrane stability. In leaves of drought-stressed soybean plants BL increases photosynthesis, content of soluble sugars and proline, and activities of antioxidant enzymes [27]. Recently, BRs have raised as decisive regulators in GDT playing the role of mediators between stress and growing processes [11]. Results have shown that BR inhibit or modulate the efficiency of plant immune signaling and responses [28,29,30]. Thus, under stress conditions, BRs can cross talk among defense signaling pathways with a broad range of hormones. For example, in rice, they have been found to be the master regulators of GAs metabolism [31,32]. Also, it has been found that exogenous treatment with BRs provoked an inhibition of pattern-associated immunity (PTI) because of some issues involving the pattern-recognition receptors (PRRs) that are going to be further explained below [33,34,35].

As summarized by Eichmann and Schäfer, 2015 [9], hormones play a crucial role in the underlying conflict between the immune and growing signaling. Growing processes are favored under non-stress conditions, hormones that stimulate cell proliferation, elongation and differentiation are secreted. Otherwise, biotic stress conditions set up fitness towards the preparation of a suitable immune response. Basically, growth hormones (AUX, GA, BR, CK) affect immune signaling, while typical stress hormones (SA, JA, ET, ABA) influence growth and development. Nevertheless, it is worth to mention that it is not possible to see immune and developmental processes apart from each other, but a hormone-regulated overall balance shifted to either growth or defense, as a response to environmental changes [36,37,38].

## 3. Receptors-Like Kinases at the Regulation Basis of GDT

Receptors-like kinases (RLKs) are a large family of receptor proteins that share a common architecture consisting of an intracellular kinase domain, a single membrane-spanning helix and an extracellular domain. These features allow sort of a structural diversification which make possible the perception of a huge variety of ligands in a specific manner [39,40]. Classification of RLKs based on the comparison of their domain structures [41] shows that leucine-rich repeat receptor-like kinases (LRR-RLKs) represent one of the largest groups, with over 170 genes in the Arabidopsis genome [42]. Studies suggest LRR domains are involved in protein-protein interactions and consist of repeating units of around 24 amino acids rich in leucine [43]. LRR-RLKs are involved in a variety of plant processes including defense and growth, moreover it is thought most of PRRs associate with other RLKs and function as part of multi-protein complexes at the cell surface, and looking deeper into their functions is the key for understanding GDT [44].

Plants have evolved two branches in their immune system to defend themselves against pathogen infections. The primary innate immune response is triggered by the detection of evolutionarily conserved microbial- or pathogen-associated molecular patterns (MAMPs or PAMPs, respectively) and is known as PAMP-triggered immunity (PTI). The second plant innate immune response is triggered by the recognition of specific pathogen effector proteins and is referred to as effector-triggered immunity (ETI) [45,46,47]. The perception of MAMPs or PAMS is carried out by patterns recognition receptors (PRRs), most of them are LRR-RLKs, and in these process hormones play a significant role.

Some receptors make up the first line of plant defense at molecular level and are the place where all PTI and ETI signaling processes start. *Xanthomonas oryzae* resistance 21 (Xa21) is a kinase that contains non-arginine-aspartate (non-RD) motif and LRR in the extracellular domain [48], which provides the ability to identify lipopolysaccharides (LPS) from bacteria [48]. The complex formed by Chitin Elicitor Binding Protein (CEBiP) and Chitin Elicitor Receptor Kinase (CERK) is on charge of detecting chitin from fungi cell wall. Besides, an insect bite erupts a series of volatile and peptidic hormones signaling cascades, those that tell the plant and some of its neighbors to prepare for a forthcoming attack [49].

On the other hand, there are receptors that participate in plant growth and developmental processes through the perception of endogenous hormone or peptide signaling. Phytosulfokine (PSK) receptor (PSKR) senses this sulfated pentapeptide that stimulate growth and root elongation; Clavata1 (CLV1) perceives the signaling peptide Clavata 3 (CLV3) to control shoot apical meristem maintenance; Erecta and Erecta-Like 1 (ERL1) recognize the peptides Epidermal Patterning Factor 1 (EPF1) and EPF2 to regulate stomatal development and patterning; Root Growth Factor (RGF) Receptor 1 (RGFR1), through RGFR5, perceives RGF peptides to mediate root meristem development. All these receptors belong to the LRR-RLK family and most of them use Somatic Embryogenesis Receptor Kinases (SERKs) as their co-receptors to turn on their intracellular transduction reactions [50,51,52,53,54,55].

Later, on this section we are going to focus on the interesting and intriguing relation between defense and hormone receptors sharing the same co-receptor protein.

Flagellin Sensing 2 (FLS2) is a LRR-RLK that operates as PRR for flagellin (*flg22*), which is a 22-amino acid peptide located at the N-terminus of the bacterial flagellum and a conserved microbial protein that is perceived as PAMP in plants, and even in animals [56]. FLS2 triggers the production of reactive oxygen species (ROS), the defense-related hormones ET and SA, deposition of secondary compounds such as callose and global transcriptional reprogramming involving WRKY transcription factors. The knowledge about FLS2 activity is important to establish a molecular control of its responses, avoiding unnecessary production of constitutive defenses that lead to excessive growing costs [57].

On the other hand, Brassinosteroid Insensitive 1 (BRI1) receptor, another member of the LRR-RLK family, is responsible for recognizing BRs, the growth-promoting hormones [19]. To recognize brassinolide (BL), its natural ligand, the receptor count on hydrophobic groove formed between the inner surface of the helical BRI1-LRR and a ~70-residue island domain [58,59]. A nonpolar cleft lined by nonpolar aromatic and aliphatic residues made up the binding site of BRI1, whereas hydroxyl groups form the cleft ridge. This is located on the surface of the receptor ectodomain. BL fits into the cleft via its nonpolar side and displays its hydroxyl groups towards the solvent and protein partners [60,61]. Therefore, it could be assumed that plants perceive steroid hormones at the plasma membrane and eventually stimulate gene expression through a signal transduction pathway. Binding of BRs to inactive BRI1 homodimers, induces auto-phosphorylation of its cytoplasmic kinase domain and thereby stimulates the interaction of BRI1 with another related LRR-RLK, SERK3, also called BRI-Associated Kinase 1 (BAK1) [62,63]. Two models have been proposed for BRI1/BAK1 interactions. The first suggests that BR allows BRI1 and BAK1 to interact, thereby allowing transphosphorylation of BAK1, whereas in the second model BRI1 and BAK1 form a complex that is stabilized and activated by BR [64]. Recently, in vitro analysis suggests that heteromeric associations between BRI1 and BAK1, as well as phosphorylation, are dependent on BR [65].

Regarding the relationship between BR signaling and immunity response, Albrecht et al. [28] showed a unidirectional inhibition of both the BAK1-dependent, FLS2-mediated immune response as well as a BAK1-independent immune response by BR perception through a yet unknown mechanism, suggesting that BAK1 is not rate-limiting in these pathways. In contrast, it has also been showed that over-expression of BRI1 in Arabidopsis brings about a reduction of BAK1-dependent, but not BAK1-independent, immune responses [29]. However, the synergistic interaction between BR signaling and immune response requires BAK1, which indicates the existence of a complex interplay between BR signaling and immunity responses involving BAK1.

In 2007, Heese et al. identified SERK3/BAK1 as a component of plant PTI [66]. In *Arabidopsis thaliana*, AtSERK3/BAK1, along with FLS2, forms a ligand-dependent complex that initiate the *flg22*-dependent responses. It is also required for full responses to unrelated PAMPs and for restriction of bacterial and oomycete infections. Thus, SERK3/BAK1 plays a significative role on plant-pathogen interaction and it seems to be involved in the initiation of PAMPs-associates downstream [66]. Formation of this complex suggests proximity of FLS2 and BAK1 in the plasma membrane. In addition, structural studies have demonstrated that the N-terminal of flg22 binds to the concave surface of FLS2 ectodomain, and the C-terminal region of the FLS2-bound flg22 interacts with BAK1 ectodomain. This interaction leads to the stabilization of FLS2-BAK1 dimerization, and thus FLS2-BAK1 heterodimerization is both ligand and receptor mediated. Hence, it is proven that flg22 binds first to FLS2 and then to BAK1; therefore, BAK1 acts as a co-receptor for flg22 signaling [67]. Current studies have demonstrated that FLS2 and BRI1 form protein clusters at different nano-domains of the plasma membrane, as it’s done by the Epidermal Growth Factor (EGF) in animals [68]. This spatial organization allows that immune and growth signaling platforms generate signaling specificity upon ligand perception, in addition to potential differential phosphorylation of common signaling components. Consequently, common signaling components could be employed along different and even antagonistic signal transduction routes [68].

## 4. Design and Synthesis of Molecules for the Exogenous Stimulation of Receptors

Supported on new organic synthesis methods, chemists have modified or synthesized analogues of naturals ligands looking forward plants’ response by the stimulation of a specific cell receptor. Besides, model mechanisms of action have been proposed to explain the bioactivity of existing molecules. In this quest, our group has obtained, through a many years’ work, a library of BRs analogs exhibiting activities similar to that observed for natural BRs-like hormones [69]. Between these new synthetic compounds there are steroids, steroidal-sapogenin and phytosterol derivatives, having several chemical functions such as oximes, ketones, epoxides, alcohols, and bromides [69]. On the other hand, molecular docking simulations have been carried out for most of these synthetic compounds (spirostanic derivatives **1**–**3**, Figure 1) to predict the affinity and binding modes between such ligands and the ectodomain of BRI1, as well as to identify the most active compounds [61]. This analysis suggested that the studied compounds were able to activate BRI1 receptor, and in the majority of cases with a lower binding energy than that corresponding to BL, the natural ligand [61].

More recently, the effect of short alkyl side chains and the configuration at C22 on the growth-promoting activity of a series of new brassinosteroid 24-norcholan-type and benzoylated analogs (BRs derivatives **4**–**7**, Figure 2) have been evaluated by the rice leaf inclination test using BL as positive control [70,71]. Molecular docking of compounds **4**–**7** indicated a possible interaction with BRI1 receptor and that ligand recognition could be enhanced by hydrophobic interactions of ligands with the BRI1LRR receptor and hydrogen bonding with BAK1 in the complex [71]. Anyhow, in all these cases it has been shown that synthetic analogs exhibit similar hormonal activity as natural BL. Thus, they can be used as precursors of promising biostimulant formulations to improve crops yields.

Additionally, BRs analogues obtained from deoxycholic and hyodeoxycholic acids have turned out to show excellent bioactivity in the rice lamina inclination assay [72,73]. Using a 3D-QSAR it has been possible to establish a structure-activity relationship which indicates that activity is favored by the following structural parameters: presence of hydroxyl groups at C22 and voluminous group in the end of alkyl chain; the hydroxyl group in C3 of ring A is indispensable [74]. The 3D contour maps showed that the growth promoting activity of the compounds was influenced mainly by electrostatic properties and the presence of hydrogen-bond acceptor groups [74].

External application of BRs analogues enhances plant growing, stress resistance, and defense responses. Application of the spirostanic analogue DI-31 (**3**) (Figure 2) on strawberry (*Fragaria x ananassa*) produced an increase on leaf greenness, number of leaves and stolon, and foliar area as compared to control plants. The effect on the activation of a defense response was also evaluated, and results revealed that it exerted a protective effect against *Botrytis cinerea*, the causal agent of the gray mold disease [75]. An increase in the calcium influx and an overexpression of defense-associated genes in response to treatment with DI-31 was observed, demonstrating the role of calcium in the signaling pathways induced by BRs in strawberry plants [76].

For plague protection, generally, farmers use pesticides that severely affect the environment and the integrity of ecosystem. Also, in many cases, transgenic crops capable of overexpressing defense genes that protect the plant against specific pathogen have been developed. However, in many cases such methods have led to the emergence of resistant strains which have turned out to cause a severer damage. Thus, the synthesis of compounds, mimicking the function of PAMPs, DAMPs, HAMPs, and plant defense hormones, has become a promising research field in chemical agriculture. Most of these compounds do not have a proven antimicrobial activity, but they activate plant innate immunity in local tissue (infected part) leading to transportation of the mobile defense signals to systemic (uninfected) tissue, resulting in a long-lasting resistance to a broad spectrum of pathogens. This acquired immunity is known as systemic acquired resistance (SAR) [77,78].

SA was one of the first endogenous compounds reported to cause SAR, and its acetylated derivative (ASA), well known as *Aspirin*, showed similar activity in tomato, accompanied by accumulation of Pathogenesis-Related (PR) proteins and resistance to Tobacco Mosaic Virus (TMV) [79]. Similar induced activities against the same virus were reported for chlorinated derivatives [80]. Structure-activity studies have shown that derivatives with electron-withdrawing groups, at **3** and **5** positions, are considerably more active as SA-like SAR stimulators [81]. More recently, a new class of SA-glycoconjugates containing hydrazide and hydrazone moieties showed significant in vivo antifungal activity in cucumber plants [82]. These conjugates induce expression of JA marker genes instead of SA marker genes. This result suggests that SA hydrazine derivative may not be a SA agonist and function through targeting of other immune signaling components [82].

Jasmonic acid and its related derivatives, including hydroxylated and JA conjugates, are hormones found all over the plant kingdom acting in numerous processes related to stress and development. The occurrence, biosynthesis and metabolism of these compounds have been reviewed [83,84]. Methyl jasmonate (MeJA), 8-((1S,2S)-3-oxo-2-((Z)-pent-2-en-1-yl) cyclopentyl) octanoic acid (OPC-8:0) (**8**)—a JA precursor—and N-jasmonoyl-L-isoleucine (JA-Ile) are among the best characterized members of the JAs family (Figure 3). Many synthetic derivatives have been obtained and their biological activity as regulators of physiological processes have been assessed [85]. For example, it has been shown that a fluorinated analog 7F-OPC-8:0 (**9**) behaves similarly to the endogenous oxylipin **8** in *A. thaliana* plants, and it also may act as systemic signal between distal tissues [86,87]. On the other hand, two diastereomeric macrolactones derived from JA-Ile were synthesized (**10** and **11**, Figure 3), and it was proved that **10** and **11** induce nicotine accumulation in *Nicotiana attenuata* leaves at similar extent as MeJA does [88]. Additionally, it was found that (**10** and **11**) activate jasmonic responsive gene (JRG) expression similarly to JA-Ile in *A. thaliana* [88]. This result indicates that the free carboxyl group of JA-Ile is not essential for this molecule to induce JRG expression [85,86]. Similar outcomes were observed in other studies, so systematic structural modifications of JA revealed the minimal structural requisites required for its bioactivity allowing for the synthesis of JA-mimics [85,89,90].

AUXs could be called the “Master of Plant Growth” and are one of the most widely studied plant hormones. There are four natural auxins that are synthesized by plants: indole-3-acetic acid (IAA), indole-3-butyric acid (IBA), 4-chloroindole-3-acetic acid (4-Cl-IAA), and phenylacetic acid (PAA) [89,90] (Figure 4).

However, at high doses the synthetic auxins are phytotoxic, inducing widespread over-reaction to auxin stimulation, which leads to injury and death [91]. Moreover, the interest in developing AUX-like compounds has increased for plant growth and weed control. Several in silico studies have noted that the transport inhibitor response 1 (TIR1) receptor family is variously selective to different auxin chemical scaffolds. Thus there exist a huge array of synthetic analogues of auxins, for instance halogenated derivatives, pyridine derivatives, quinolones, aromatic acetates, benzoic acids, and phenoxyacetic acids [92].

Therefore, exist a variety of synthetic molecules that stimulate both growth and defense by binding to cell receptors and mimicking the functions of natural hormones, so they might be able to trigger a response, harsher or milder, depending on what is intended to. Synthetic polymers, chitosan and others carbohydrate derivatives, lipoproteins, vitamin-like molecules, cyclopentanoic acids, even antimicrobial peptides, uracil and urea derivatives, and still mineral salts have been used because of their bioactivities under specific conditions [93]. Despite these advantages, it is worth to consider that the activity of almost every reported molecule is restrained to in vitro conditions or to a small number of plant species. Other potential concern related with synthetic molecules deal with the indiscriminate use of chemicals, which is responsible for the majority of soil contamination problems, although molecules from natural sources and their derivatives have proven to have a great bioactivity and be innocuous to the environment.

## 5. GDT Influence over Secondary Metabolism

Trade-off*s* take part in the secondary metabolism as well, and hence depending upon the conditions and features of the environment where the plant is in, the nature and concentration of secondary metabolites change. Secondary metabolites are necessary for a ubiquitous amount of processes that involve transduction, cell adaptation and survival, communication, and protection, but the role of secondary metabolism upon defense and defense response has been comprehensively studied in the last decade [94]. Transcription factors responsible for regulating the expression of secondary metabolites biosynthesis genes, are the result at molecular level of a response that was first activated by an intra- or extracellular signal perceived by the membrane receptors, in a similar way as explained above in this review [95].

The concentration of secondary metabolites could serve as reference to know about the development stages and conditions of the ecosystem that surrounds the plant organism: instances of nutrients availability, stress condition, herbivore attack, and some others. The accumulation and cell secretion of nicotine and caffeine in tobacco and coffee plants, respectively, upon tissue wounding, that could be interpreted as herbivore attack, could explain the insecticide activity of these compounds [96,97]. The high conjugation displayed in anthocyanins’ structure allow them to interact with light and absorb some kinds of radiations, thus its production is stimulated under high-light exposure. As they also are really good antioxidants, conditions such as nutrients deficiency, and pathogen attack and wounding, may induce an over production of anthocyanins [98,99]. In leaves and stem of *Tithonia diversifolia*, the production of sesquiterpenes, lactones as tagitinin A and C (tagitinin C with potent cytotoxic activity on human malignant cells [100]) and phenolics as 5-*O*-caffeoylquinic acid (Figure 5) is determined by the amount of rainfall and temperature changes occurring with seasonal variation [101]. On the other hand, the production of phenolics and subsequent incorporation in plant cell wall as suberin or lignin is increased by cold stress [94], while tree adaptation was associated with production of chlorogenic acid at high levels [95].

Some secondary metabolites are volatile organic compounds (VOCs), so they can spread along and establish a sort of interaction within the plant itself and with its similar ones on the field-side. Moreover, it was recently proposed that these compounds may work in synergy with other secondary metabolites and hormones which are all synthesized to regulate senescence [102]. For instance, the synergism between the production of isoprenoids and biosynthesis of cytokinins turns out in increased antioxidant activity at the foliar level, which by preventing degradation processes prolongs life span of leaves and flowers. Trade-offs between benefits and costs of VOC emission as stress relief compounds are still not well understood as experiments with transgenic plants suggest that the metabolic cost for emitting isoprene (the most abundant VOC released from leaves) outweighs benefits [103,104].

Many ecological experiments have been designed with the aim of observing behavior of plants in wildlife conditions and get a comprehensive approach to the trade-off hypothesis. As an example, *Stryphnodendron adstringens* is a tree whose bark is used in Brazil as a source of medicinal tannins, nevertheless after bark harvesting, the tree generally dies. Tuller et al. in 2018 [105], in a manipulative field experiment, tested the hypothesis that harvesting leaves, which might serve as an alternative source of tannin, would be less detrimental for tree survival, growth, reproduction, and defense. Clipping the totality of leaves induced a trade-off such that reproduction (number of fruits) decreased but tannin concentration increased in plant tissues, as leaves were the highest defended tissue. These results are consistent with the hypothesis that damage induces higher secondary compound production to reduce the likelihood of subsequent herbivore damage [106]. In addition, investment in defensive compounds following herbivore attack instead of reproduction may be more profitable for long-lived trees, like *S. adstringens*, because they may have many future reproductive episodes [105].

## 6. Trade-Offs Lead to Physiological Changes upon Environmental Stimuli

Until now, we have dealt only with the basis of trade-off*s* at a molecular level, but we cannot forget that all signaling, and responses start with an environmental stimulus. The huge array of cell receptors has evolved to become the way as cells perceive their surroundings and interact with it. Then, once the stimuli are sensed and the responses given, what comes next? Well, for plants, as for other living organisms, is quite important to know if the given response was effective, so feedback processes must be established. Therefore, a changing environment is not healthy for plant development since it makes plants more susceptible to biotic and abiotic stresses. Environmental factors are also responsible of phenotype-shaping whose expression depends on the genes that have been activated by the transduced signal.

Temperature is one of the main factors intervening in tolerance responses to stress conditions, because such factor is always fluctuating along with the day, and changes abruptly with the seasons of the year. Furthermore, global climate change is showing a tendency to increasing temperature, and the understanding of the impact of this phenomenon on plants is a priority to mitigate the contemporaneous agriculture problems. In plants, temperature sensing is associated with fluctuations in membrane fluidity, histone modifications, activation of protein kinase cascades, and generation of ROS [107].

Rice plants infected with *Xanthomonas oryzae*, at elevated temperature, displayed upregulation of ABA biosynthesis and signaling genes, and downregulation of SA-responsive genes. Although, under the same conditions, rice plant carrying the *Xa7* gene exhibit resistance to this pathogen, suggesting that SA-independent defense signaling occurs at high temperature [108]. This fact shows that temperature is involved in the regulation of resistance genes expression upon pathogen attack and hormone crosstalk, since temperature accelerates breakdown in some plant-pathogen systems, though in some other cases it stimulates an enhanced response [2,109] Similar behavior was observed in *Arabidopsis* exposed to *P. syringae*, where susceptibility was enhanced by increasing temperature. This effect is due to inhibition of SA-biosynthetic and SA-responsive gene expression, and upregulating genes involved in JA-mediated signaling and ABA biosynthesis, suggesting that an important interplay between SA and ABA/JA signaling is operating [110].

Extremes conditions of temperature affect negatively both the growth and defense processes as plants have to adequate transcriptional pathways and produce osmotic factors to avoid freezing. Cold stress stimulates membrane rigidification and cytoskeletal rearrangement. The cold signal is perceived by the membrane and temperature sensor (COLD1/RGA1) and other components, leading to an influx of Ca^2+^, ROS production, ABA accumulation, and MAPK cascade (OsMKK6-OsMPK3) reactions [111]. Several studies lead us to believe that ROS are the key for understanding trade-offs under stress conditions, since at low concentration they trigger defenses and developmental responses at early stages, whereas at highest concentration they attack the cell membrane for destroying the cell under breakdown of the defense barrier [112,113]. The redirection of functions upon chilling stress is logically associated with the DNA damage that induces protective death of columella stem cell daughters leading to restoration of the aux-in maximum in the quiescent center (QC) and maintenance of functional stem cell niche activity in *Arabidopsis* roots. This breakthrough supports a new concept in which DNA damage and ROS production coordinate cell fate during responses to chilling stress [111,114].

Soil salinity is nowadays one of the most extended problems that agriculture is facing worldwide, it has been provoked by mankind irresponsible soil exploitation, and bad practices applied to extensive agriculture. This issue provokes significant economic losses because of its effects on plant metabolism, it causes a total disorder of growth regulators and uncoupling major physiological and biochemical processes [115]. Under salt stress the secretion of ET occurs, as it does under other types of stresses. Through ET burst, plants alert to other parts of itself and to its similar ones that it is under a harmful damage, so it can start up with stress responses. However, when the stress persists, it can trigger the initiation of senescence, chlorosis, and abscission processes, and ultimately lead to plant death [116]. Moreover, a recent research showed that lettuce plants exogenously treated with BRs and spirostanic synthetic analogs, reduced significantly the ET production in roots and shoots [117]. Furthermore, it has been shown that rice’s seedlings growth under NaCl stress is significantly enhanced by treatment with BRs and its synthetic analogs [118,119,120,121]. Also pigment concentration increases and proline content decreases in leaves, as an evidence of adaptability to such stress conditions [33,69]. As BRs are so environmentally friendly, this result is an evidence of sustainable solutions to the salinity problems that affect almost the whole world today.

Biotic stress causes numerous losses in crops yearly. Both above- and belowground microbes may affect the growth-defense balance of plants. In ecology, the growth-defense hypothesis states that the faster the plant grows the greater the damages due to pathogen attacks are, and vice versa. Fast-growing species invest most of the available resources in plant growth rather than in defense mechanisms, whereas the opposite occurs in slower-growing species [122]. The majority of the tests designed to prove this hypothesis have been carried out with aboveground tissues and this behavior seems to be the norm [5,122,123,124,125]. Similar results have been reported for the existing relationship between belowground growth rate and soil biota effects (microbial fungi and bacteria) [101]. However, the established relationships suggests that belowground plants are involved on many interactions via their roots with antagonistic and mutualistic organisms [101]. Consequently, slower-growing species are benefitted from the soil mutualist at higher extent than faster-growing species. These studies demonstrate that GDT could be a complex regulator phenomenon which can be used as a predictor of plant community responses to pathogens.

An interesting fact, evidencing GDT-induced changes is that axial resin ducts, which are costly defensive structures, remain imprinted in tree rings of conifers, so they might become a valuable proxy to evaluate defensive investment. Vázquez-González et al. [126] recently studied the responses to both spatial and temporal environmental variation in resin duct production, and to explore growth-defense trade-offs. To that aim, they applied dendrochronological procedures to quantify annual growth and resin duct production during a 31-year-period in a Mediterranean pine species, including trees from nine populations planted in two common gardens. Interestingly they found that annual resin duct production differed among populations of *Pinus pinaster*. Such variation in defensive traits can be logically expected as a proof of the optimization of resources provided by the ecosystem which is part of the processes for long-term evolutionary adaptation. Besides, they found a strong evidence of a physiological growth-defense trade-off at the phenotypic level, indicated by a negative correlation between annual basal area increment—feature associated with growth—and annual resin duct density. Climate conditions are strongly related with the above correlation since it is expected that growth-defense trade-offs are more likely to emerge under limiting environmental conditions determining physiological constrains. Populations that evolved under more favorable growth conditions showed stronger physiological defensive constraints. This can be explained by the fact that such populations were the most dissimilar in climate conditions to the sites where the trees were planted.

## 7. Conclusions and Perspectives

As plants either grow or defend themselves, they have to stablish a still not completely known balance to warrant their survival in the location where they are fixed. Everyday soil exploitation and bad practices combined with lack of nutrients, salinity, drought, degradation, along with the climate change, which effect is nowadays increasingly evident, are the main challenges that demands from us a deeper knowledge of what really occurs inside the plant organism and how they express to simple view. With this review we have summarized the growth-defense trade-offs in a comprehensive way, intended to trace a guiding line starting with the processes at the molecular level leading to phenotypic changes that plants can experience upon their interaction with the environment. The understanding of these topics is of great importance for today’s plant scientists and agricultural producers, since sustainable technologies aimed to satisfy and ensure the necessities the humankind demands are required to be developed. Therefore, eco-friendly solutions to help plants to get a better adaptation to stress conditions and achieve better production yields are a great challenge and a must for plant science research.

## Figures and Tables

**Figure 1 ijms-22-00693-f001:**
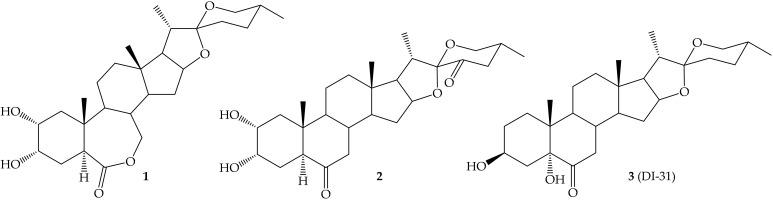
Spirostanic derivatives **1**–**3**.

**Figure 2 ijms-22-00693-f002:**
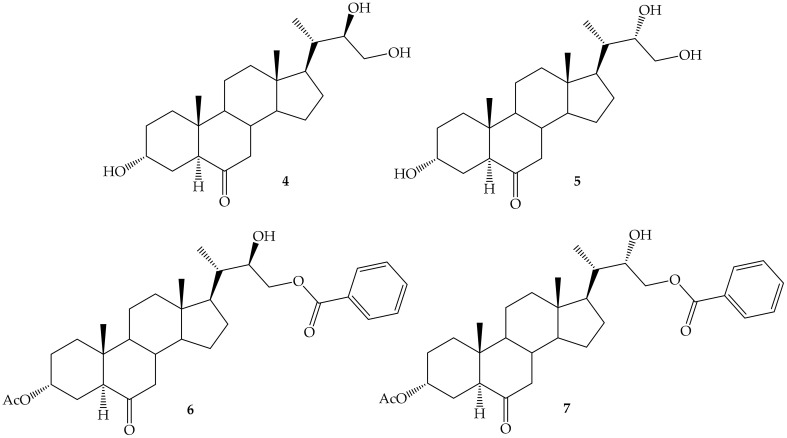
Brassinosteroid 24-norcholan-type and benzoylated analogs.

**Figure 3 ijms-22-00693-f003:**
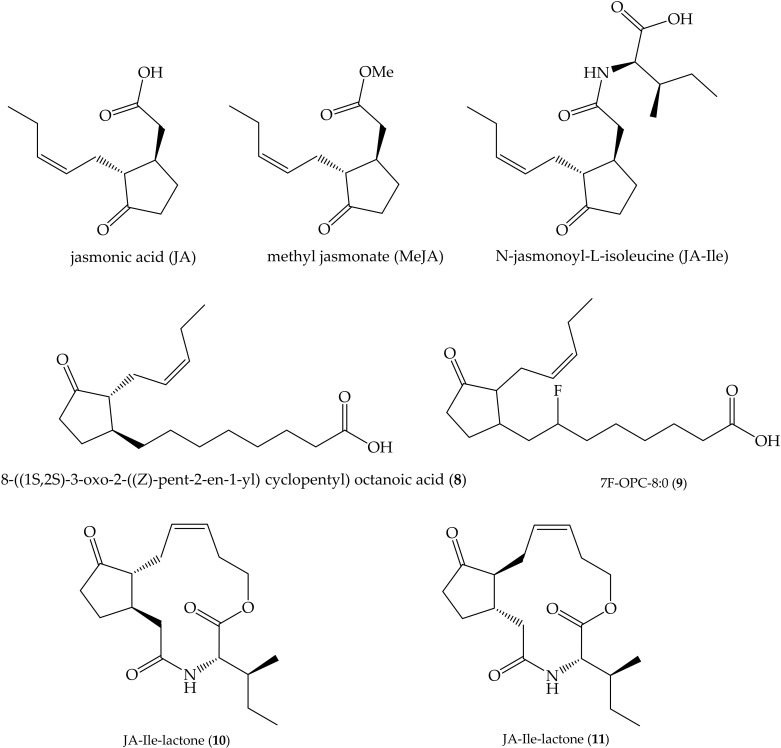
Jasmonic acid (JA), natural derivatives methyl jasmonate (MeJA), N-jasmonoyl-L-isoleucine (JA-Ile), JA precursor 8-((1S,2S)-3-oxo-2-((Z)-pent-2-en-1-yl) cyclopentyl) octanoic acid OPC-8:0 (**8**), and synthetic analogs 7F-OPC-8:0 (**9**) and JA-Ile macrolactones (**10** and **11**).

**Figure 4 ijms-22-00693-f004:**
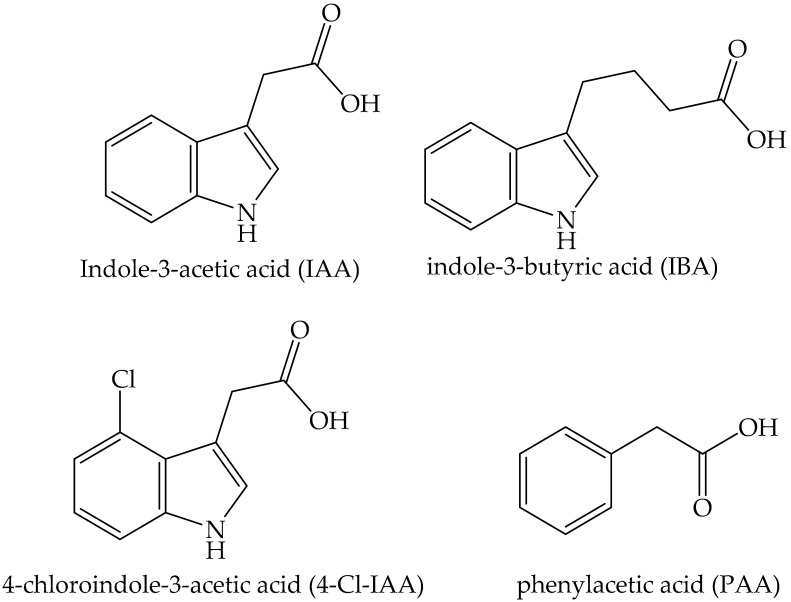
Chemical structures of natural auxins.

**Figure 5 ijms-22-00693-f005:**
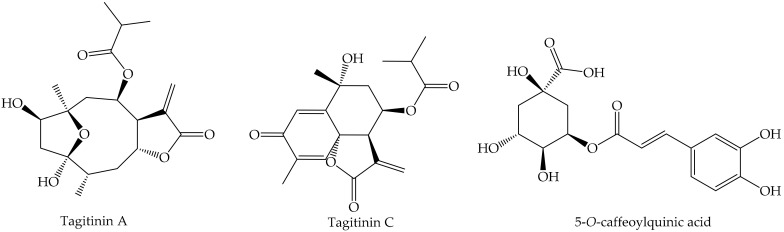
Chemical structures of tagitinin A, C and 5-*O*-caffeoylquinic acid detected in *T. diversifolia* extracts.

## Data Availability

The data presented in this study are available on request from the corresponding author.

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
