# Peer review of "Plant Growth-Defense Trade-Offs: Molecular Processes Leading to Physiological Changes"

_ijms, 2021, doi:10.3390/ijms22020693_

Round 1
Reviewer 1 Report
Authors tried to summarize the growth-defense trade-offs starting with the processes at the molecular level leading to phenotypic changes that plants can experience upon their interaction with the environment. However, most molecular parts are solely describing hormonal controls, it will be more informative to include key regulator in the pathways, especially those negative regulators of biotic and abiotic stresses which suppress defense responses in normal condition to keep plant growth and how these regulators modulated by phytohormones under stress conditions.
Other comments:
P1 line 35 “drought” instead of “draught”?
It is odd to term section 2 as “Result”. Please summarize section 2 content in a sentence as Section title!
Figure 1 is unnecessary, as it is easy to be found online. It is better to elucidate more about the content in Section 2.1, e.g. genes responsive to phytohormones are involved to defense and / or growth and presented in form of table.
- 3 Paragraph started from line 87: content is not in details. Not much information got from this paragraph. Only deliver message that “hormones are involved”, please at least state clearly which hormone participates in which mechanism!
- 3 Paragraph started from line96: As BL increment leads to increase in growth and stress responses, then do you mean it is the exception of GDT?
Figure 4 is totally copied from original reference (Ferrer-Pertuz, et al. 2017 Figure 6). It is really not good even though you cited the reference. Please aware of the issue of plagiarism.
Most points in Section 2.4 is describing effects of abiotic stresses on growth and biotic stresses, it is not accurate to entitle it as “Environmental factors are the external regulators of trade-offs “. This title is not precise.
- 10 line384 Please cite references!
- 11 line 390 Better to have new section for secondary metabolite! It should not under the section of “Environmental factors are the external regulators of trade-offs “
Author Response
Authors tried to summarize the growth-defense trade-offs starting with the processes at the molecular level leading to phenotypic changes that plants can experience upon their interaction with the environment. However, most molecular parts are solely describing hormonal controls, it will be more informative to include key regulator in the pathways, especially those negative regulators of biotic and abiotic stresses which suppress defense responses in normal condition to keep plant growth and how these regulators modulated by phytohormones under stress conditions.
Other comments:
P1 line 35 “drought” instead of “draught”?
The typo has been corrected
It is odd to term section 2 as “Result”. Please summarize section 2 content in a sentence as Section title!
The title “Results” has been eliminated
Figure 1 is unnecessary, as it is easy to be found online. It is better to elucidate more about the content in Section 2.1, e.g. genes responsive to phytohormones are involved to defense and / or growth and presented in form of table.
We believe that Figure 1 is useful for the presentation and discussion of data because these hormones are named more than once in the text.
- 3 Paragraph started from line 87: content is not in details. Not much information got from this paragraph. Only deliver message that “hormones are involved”, please at least state clearly which hormone participates in which mechanism!
This paragraph was relocated and information about the involved hormones was included. Lines 105-113
- 3 Paragraph started from line96: As BL increment leads to increase in growth and stress responses, then do you mean it is the exception of GDT?
This sentence was modified to make clear that growing effect of BL is on Soybean leaves submitted to drought stress. Now it reads “In leaves of drought-stressed soybean plants BL increases photosynthesis,……..”
Figure 4 is totally copied from original reference (Ferrer-Pertuz, et al. 2017 Figure 6). It is really not good even though you cited the reference. Please aware of the issue of plagiarism.
Figure 4 was used because it belongs to an article of L. Espinoza, corresponding author, published in this Journal.
Most points in Section 2.4 is describing effects of abiotic stresses on growth and biotic stresses, it is not accurate to entitle it as “Environmental factors are the external regulators of trade-offs “. This title is not precise.
The title was changed. Now it reads “GDT influence over secondary metabolism”
- 10 line384 Please cite references!
Four new references were added
- 11 line 390 Better to have new section for secondary metabolite! It should not under the section of “Environmental factors are the external regulators of trade-offs “
This is a very good point and therefore we have split section 2.4 in two sections, namely the current 2.4 about secondary metabolites and 2.5 about physiological effects. The text was kept the same since we considered it was enough to form two independent sections
Reviewer 2 Report
The MS of Figueroa-Macías et al, develops the question of plant growth and defense trade-offs, focusing on the different molecular and physiological responses of plants triggered by external and internal stimuli. Overall, MS is interesting and brings together several examples of the strategies that plants adopt to ensure a balance between defense and growth.
The organization of the MS is not clear and the titles chosen for the different chapters are rather strange and do not describe the content.
In my opinion some parts of MS can be eliminated as mere descriptions of known concepts (beginning of chapters 2.3, 2.2).
Some points:
The title should be changed. To me it is not clear what the meanings of the sentence "from the bottom to the top ".
Line 54: "Results" should be erased.
Line 113-121: can be cancelled, this part is not necessary
Line 211-219: can be cancelled, this part is not necessary
Line 350-352: something is lacking in the sentence so it is not clear the connection between pathogen defence and temperature.
Author Response
The MS of Figueroa-Macías et al, develops the question of plant growth and defense trade-offs, focusing on the different molecular and physiological responses of plants triggered by external and internal stimuli. Overall, MS is interesting and brings together several examples of the strategies that plants adopt to ensure a balance between defense and growth.
The organization of the MS is not clear and the titles chosen for the different chapters are rather strange and do not describe the content.
All section headings have been changed so now they describe the following content more properly
In my opinion some parts of MS can be eliminated as mere descriptions of known concepts (beginning of chapters 2.3, 2.2).
These paragraphs were eliminated
Some points:
The title should be changed. To me it is not clear what the meanings of the sentence "from the bottom to the top ".
The title was changed and now it reads “Plant growth-defense trade-offs: molecular processes leading to physiological changes”
Line 54: "Results" should be erased.
This title was eliminated
Line 113-121: can be cancelled, this part is not necessary
The whole paragraph was eliminated
Line 211-219: can be cancelled, this part is not necessary
The whole paragraph was eliminated
Line 350-352: something is lacking in the sentence so it is not clear the connection between pathogen defence and temperature.
The paragraph was restructured and the relation between temperature and defense is more clearly established
Round 2
Reviewer 1 Report
With respect to the revised manuscript, some points previously mentioned have been revised appropriately. However, some points are still not respond well or even left unanswered.
Point 1: For Figure 1, even these hormones are named more than once in the text, their structures for function are not the points being discussed in the text and these information could easily be found online. If authors really insists to keep the figure, please add content to discuss how these hormone structure related to their functions!!
Point 2:Authors have not respond to this point: "It is better to elucidate more about the content in Section 2.1, e.g. genes responsive to phytohormones are involved to defense and / or growth and presented in form of table."
Point 3: It is better to elaborate more about the content of newly added Lines 105-113.
Point 4: For Figure 4, even the figure copied from the publication of the same corresponding author, it is still not good to have nearly the same figure in two publications, when there is no new information integrated to the figure, just citing the original reference is enough.
Author Response
With respect to the revised manuscript, some points previously mentioned have been revised appropriately. However, some points are still not respond well or even left unanswered.
Point 1: For Figure 1, even these hormones are named more than once in the text, their structures for function are not the points being discussed in the text and these information could easily be found online. If authors really insists to keep the figure, please add content to discuss how these hormone structure related to their functions!!
We have decided to remove this figure
Point 2:Authors have not respond to this point: "It is better to elucidate more about the content in Section 2.1, e.g. genes responsive to phytohormones are involved to defense and / or growth and presented in form of table."
We have added a new paragraph explaining the roles of phytohormones in regulating both the immune and growth plant responses (lines 86-95). Herein we mention some of the genes that are involved in these responses. However, we have not made an exhaustive search of genes that are up-or downregulated by these hormones. We believe that this is a task that it is worth to be made but it is well beyond the scope of this review.
Point 3: It is better to elaborate more about the content of newly added Lines 105-113.
We have added a phrase explaining our statement that BRs are important in GDT with two new references. The mechanism is explained in section 2.2.
Point 4: For Figure 4, even the figure copied from the publication of the same corresponding author, it is still not good to have nearly the same figure in two publications, when there is no new information integrated to the figure, just citing the original reference is enough.
We have eliminated figure 4 and cited the results in the text. The new text is highlighted
Round 3
Reviewer 1 Report
Authors reponded and revised the manuscript nearly what I requested.